# Family Attachment, Sexuality, and Sexual Recidivism in a Sample of Italian Sexual Offenders: Preliminary Results

**DOI:** 10.3390/healthcare11111586

**Published:** 2023-05-29

**Authors:** Valeria Saladino, Stefano Eleuteri, Angela Nuzzi, Valeria Verrastro

**Affiliations:** 1Department of Human Sciences, Society and Health, University of Cassino and Southern Lazio, 03043 Cassino, Italy; 2Department of Psychology, Sapienza University, 00185 Rome, Italy; 3Institute for the Study of Psychotherapies, 00185 Rome, Italy; 4Department of Health Sciences, University “Magna Graecia” of Catanzaro, 88100 Catanzaro, Italy

**Keywords:** sex offender, attachment, relapse, violence, sexual sensation-seeking

## Abstract

Objective: The research aims to investigate family communication regarding sexuality and the possible link between insecure attachment, violence in relationships, and the tendency toward sexual sensation-seeking in a sample of Italian sexual offenders. Design and method: We evaluated 29 male sexual offenders in two correctional facilities of Southern Lazio (Italy) (mean age = 40.76; SD = 11.16). The participants completed general questions about their family and sexual education and fulfilled the following questionnaires: Compulsive Sexual Behavior Inventory (CSBI), Sexual Sensation-seeking Scale (SSSS), and the High-Risk Situation Checklist, adapted in Italian, as well as the Attachment Style Questionnaire (ASQ), validated in Italian. Results: Most of the participants had never talked about sex within their family and perceived a severe or abusive education during childhood. In addition, positive correlations emerged between SSSS and the two scales of the CSBI, as well as between insecure attachment style, CSBI, and sexual sensation-seeking. The participants also reported some critical issues regarding the personal perception of high-risk situations linked to sexual relapse. Conclusions: The data suggest factors to investigate, such as family education and relationships and the personal perception of sexual recidivism. The results might be effective in treatment and prevention programs among sex offenders.

## 1. Introduction

### 1.1. Sex Talk within the Family and Attachment

The importance of family communication regarding sex to the prevention of risky behaviors in young adults has been highlighted. Parents are the primary source from which children create their attitudes toward sex, and certain negative family communication patterns are considered predictors of perceptions of sexual communication as threatening [1]. Yoder et al. [2], in a study on youth sexual offenders, found a relationship between poor communication with their mothers and the severity of the sexual offense. The authors related more communication and interaction with peers involved in criminality with the perpetration of severe sexual offenses. In addition, youth sexual offenders, relative to non-sexual offenders, had more problems with maternal attachment and in all areas of paternal attachment, except communication. Lower trust with their mother has been associated with more victims, whereas lower trust with their mother and higher alienation with their father were associated with non-sexual offending. Youths who have committed sexual crimes reported higher stress scores and more problems in their family relationships.

The family system has an important role and impacts on the risk factors that increase sexual criminality and recidivism, including a negative communication pattern between parents and child, attachment deficits, internal conflict, and sexual trauma histories [2]. Attachment style deficits have been described in both sexual and non-sexual offenders. Negative experiences in the context of early attachment relationships can contribute to a limited capacity to trust others and to developing risky behaviors [3]. For instance, childhood trauma can lead to an insecure attachment associated with criminal behaviors [4]. Zaniewski et al. [5] found that insecure attachment strategies and unresolved traumas or losses were linked to harmful sexual behaviors in young men and their parents. Father–child attachment could impact the son’s involvement in sexual violence. For instance, the witnessing of acts of sexual abuse from fathers in the family context influences sexual offending behavior [6].

Grady et al. [7] investigated the relationships between experiencing abuse or trauma, attachment styles (anxious-avoidant and anxious-ambivalent), and the risk factors associated with sexual offending, including regulation deficits. Grady and Shields [8] previously demonstrated that sex offenders with anxious attachment present more deficits in emotion regulation. The authors found a direct relationship between physical abuse and both attachment styles. In addition, the anxious-ambivalent and anxious-avoidant attachment styles related similarly to regulation deficits. However, the experience of trauma only predicted regulation deficits for anxious-ambivalent attachment, but not in the anxious-avoidant attachment model. Men who had experienced childhood abuse were more likely to develop an insecure attachment. This attachment is related to impulsivity and self-regulation problems—such as emotional, cognitive, and behavioral issues—that are considered significant components of a healthy relationship [9]. Violent experiences can influence the internal models adopted in future relationships, including violence and coercion. For this reason, insecure attachments among sex offenders compromise their regulation skills, leading them to react through coercive, violent, or deviant sexual behavior [10].

### 1.2. Compulsive Sexuality and Sexual Sensation-Seeking

Impulsivity and sensation-seeking have been frequently related to criminal conduct and risky behavior. Regarding sexual behavior, impulsivity is characterized by a lack of control and a tendency to act without foresight, whereas sexual sensation-seeking is defined as the propensity to engage in novel or intense sexual activities. Sensation-seeking and impulsivity have also been associated with Compulsive Sexual Behavior Disorder (CSBD), an impulse control disorder recently introduced in the ICD-11 and characterized by a difficulty to control sexual behavior [11]. However, there is little evidence on the relationship between sexual offending, sensation-seeking, and impulsivity.

Ryan et al. [12] compared impulsivity and compulsivity among different groups of sexual offenders and general offenders. Although general offenders scored lower on the measures of sexual compulsivity and sexual sensation-seeking, the scores of general impulsivity and sensation-seeking were not statistically different across the groups. This result was in line with Perley-Robertson et al.’s [13] data on impulsivity among offenders, including sex offenders.

According to recent studies, impulsivity and sensation-seeking have been found to be more linked with CSBD and less with sexual offenses. Efrati et al. [14,15] reported lower rates of CSBD among sex offenders in contrast with Sexaholics Anonymous (SA). These results seem to be linked to differences in maladaptive schemas about the self and others, together with impulsivity and sensation-seeking. 

### 1.3. Sexual Recidivism

The existing literature on sexual recidivism rates is not consistent. This aspect is related to the methodology adopted, the length of the follow-up period, the type of sex offense, and the definition of recidivism. Moreover, many episodes of sexual offenses are unreported. Therefore, the rates of relapse could be underestimated. A recent systematic review suggested that older sex offenders have lower rates of sexual recidivism than young offenders, who have a major risk of reoffending [16]. In particular, the risk of recidivism seems to be related to social factors and the quality of relationships [17]. According to Ozkan et al. [18], sex offense recidivism is best predicted by two or more prior sex offense convictions. In addition, altered family factors, including violent parent-child relationships and deficient attachment, seem to be involved in further sexual recidivism and should be considered in the analysis of recidivism.

Little is known about the perception that offenders have about recidivism. This aspect could be central in developing relapse prevention programs that consider the specific profile of individuals and not just their offenses.

This brief research report aims to report the collected data on a sample of Italian sexual offenders, investigating the connection between family communication about sexuality, insecure attachment, violence in relationships, and the tendency to sexual sensation-seeking. In addition, we are interested in evaluating the personal perception of the participants on the risk factors that might lead to sexual recidivism.

## 2. Methods

### 2.1. Participants and Procedures

The sample is composed of 29 male sex offenders (mean age = 40.76; SD = 11.16; age range: 21–74), randomly recruited from two correctional facilities in Southern Lazio, according to the following inclusion criteria: (a) participants have received a final sentence for sexual offense (for privacy reasons we did not distinguish between abuse against minors or women, nor specify the offense); (b) participants should be male; (c) participants should be Italian; (d) participant should be aged 18+.

Foreign and judging participants were excluded from the study.

The Italian Ministry of Justice and the Department of Penitentiary Administration authorized the research. Secondly, the researchers explained the aim of the research to participants and obtained informed consent from the latter. The study was conducted in accordance with the Declaration of Helsinki and approved in January 2019 by the Institutional Review Board of the Institute for the Study of Psychotherapies, School of Specialization in Brief Psychotherapies with a Strategic Approach. The ethical approval code is ISP-IRB-2019-2.

Questionnaires were administered under the supervision of police officers of the correctional facilities and the research team. Compilation took approximately 60 min. Participants were informed of the complete anonymity of the information they were about to provide. Participants were aware that they could also stop completing the questionnaire at any time and request that the information they provided not be considered.

### 2.2. Measures

Socio-demographic Questionnaire structured by the researchers of the University of Cassino and Southern Lazio [19], this questionnaire investigates the anamnestic situation of the person, considering the family and social profile, the involvement in risky behaviors, substances abuse, accidents, traumas, and problematic family histories and relationships. The Socio-demographic Questionnaire also measures the perceived parental discipline, which evaluates the quality of the discipline received by parents in terms of severity, such as physical punishments or psychological humiliation, or feeling of not being loved, examining the personal perception of having received a harsh or abusive discipline in childhood (Would you say that you have received a severe discipline during childhood? For instance, did your parents physically punish you, humiliate you, or did not make you feel loved?).

Attachment Style Questionnaire (ASQ) [20,21] is a self-report composed of 40 items on a Likert scale from 1 to 6 (1 = totally disagree; 6 = totally agree) and evaluates family relationships and attachment. This measure includes five factors. The Confidence scale (ASQ-F1) represents secure attachment: “I trust I can rely on others in times of need” (Cronbach alpha of 0.66). The Discomfort with Closeness scale (ASQ-F2), “For me, it is problematic to depend on others” (Cronbach alpha of 0.49), and the Relationships as Secondary scale (ASQ-F3), “Engaging in one’s activities is more important than building good relationships” (Cronbach alpha of 0.55), describe the insecure avoiding/detached style. The Need for Approval scale (ASQ-F4), “For me, it is very important to be pleasant to others” (Cronbach alpha of 0.70), and the Preoccupation with Relationships scale (ASQ-F5) “I care a lot about my relationships” (Cronbach alpha of 0.76), represent the preoccupied or anxious/ambivalent style. The ASQ is interpreted according to the theoretic framework of the cognitive schemas associated with the dimensions of the attachment theory, considering the relationship of each scale with a specific type of attachment, as described above (See Appendix A for the Italian and English Version).

Compulsive Sexual Behavior Inventory (CSBI) [22,23] is a self-report adapted in Italian by the researchers of the University of Cassino and Southern Lazio. The questionnaire evaluates compulsive sexual behavior in 22 items on a Likert scale from 1 to 5 (1 = never; 5 = very frequently). CSBI considers recurrent sexual urges and fantasies that may lead to paraphilic sexual behaviors. Higher scores indicate more CSB. The questionnaire includes two factors. The control on sexual behavior scale (item 1–13), “How often have you had trouble controlling your sexual urges?” (Cronbach alpha of 0.70), and the use of violence in relationships scale (item 14–22), “Have you given others physical pain for sexual pleasure?” (Cronbach alpha of 0.86). The sum of the two scales composes the total CSBI score. (Link to the English version https://link.springer.com/article/10.1007/s10508-006-9127-2/tables/2, accessed on 18 May 2023. See Appendix A for the Italian Version).

Sexual Sensation-seeking Scale (SSSS) [24,25] is a self-report adapted in Italian by the researchers of the University of Cassino and Southern Lazio. The questionnaire includes 10 items on a Likert scale from 1 to 4 (1 = not at all like me; 4 = very much like me). The SSSS evaluates the sensation-seeking construct, derived from Zuckerman’s [26] Sensation-seeking Scale, with items redefined for sexuality, including varied or novel sexual experiences: “I like wild and uninhibited sexual encounters”; “I like to have new and exciting sexual experiences and sensations”. Higher scores indicate more SSS. Scoring consists of summing all the items to calculate the global tendency of sexual sensation-seeking and take the mean response (sum of items/10). The questionnaire has a Cronbach alpha of 0.66. (Link to the English version https://scales.arabpsychology.com/s/sexual-sensation-seeking-scale/, accessed on 18 May 2023. See Appendix A for the Italian Version).

High-Risk Situation Checklist developed by David M. Price [23,27] for sexual offenses and adapted and translated into Italian by researchers from the University of Cassino and Southern Lazio. The High-Risk Situation Checklist evaluates the perception of the risk related to recidivism. The measure evaluates the emotional, social, situational, and possible treatment elements. It consists of six categories that investigate the factors connected to the hypothetical possibility to relapse: (1) negative emotions (14 items), “Which of these negative emotions could lead you to reiterate the offense?”; (2) positive emotions (11 items), “Which of these positive emotions could lead you to reiterate the offense?”; (3) thoughts and actions (18 items), “Which of these thoughts and actions could lead you to reiterate the offense?”; (4) characteristics of the environment (11 items), “Which of these characteristics of the environment could lead you to reiterate the offense?”; (5) rehabilitation programs, “What reasons could negatively affect the success of possible rehabilitation programs?” (9 items); and (6) other positive or negative situations (8 items), “What other situations or happenings could change your behavior?”. Participants identify one item on the checklist for each category. The risk factor related to the category is evaluated according to the frequency of item selection. This evaluation derives from an individual perception of the risk of each item. This measurement could be an important tool for clinicians and psychologists to evaluate and establish possible strategies for preventing critical issues related to sexual offenders. (Link to the English version https://www.tandfonline.com/doi/abs/10.1080/10720169908400193, accessed on 18 May 2023. See Appendix A for the Italian Version).

### 2.3. Statistical Analysis

The data were analyzed using the Statistical Package for Social Sciences (Version 26.0, SPSS Inc., Armonk, NY, USA) [28]. To explore family education and relationships among sexual offenders, descriptive analysis was used, while to investigate the association between insecure attachment, the use of coercion and violence in relationships, and sexual sensation-seeking, a Pearson correlation was used.

## 3. Results

The descriptive analysis shows the attitude of the sample towards sexuality. Specifically, Table 1 reports the percentages of family communication about sexuality and the subjects’ style of education, abuses, and type of affective relationships.

From the Pearson correlation (see Table 2) emerges a positive association between the two scales of the Compulsive Sexual Behavior Inventory and the Sexual Sensation-seeking Scale. Regarding the relationship between attachment style and the other variables, we analyzed the five factors of the Attachment Style Questionnaire (ASQ) individually. Our analysis shows a positive relation among “Relationships as Secondary scale” (part of avoidant style), violence in relationships, and sexual sensation-seeking. In addition, the “Preoccupation with Relationships scale” (part of preoccupied or anxious/ambivalent style) positively correlates with sexual sensation-seeking and difficulty in controlling sexual impulses.

Finally, from the analysis of the High-Risk Situation Checklist (see Table 3), the sample reported: “anger and problems in managing it” under negative emotions (first category), “sense of control on the behavior” under positive emotions (second category), “my behavior is correct” under thoughts about crimes (third category), “Have contact with other offenders” under neighborhood characteristic (fourth category), “Not to be confident in the treatment” under rehabilitation programs (fifth category), and “Thinking about the future” in other general situations (sixth category).

## 4. Discussion

### Main Findings

Our study aims to explore familial sex talk, education, and attachment among a sample of 29 male sex offenders (mean age = 40.76; SD = 11.16) recruited in two correctional facilities in Southern Lazio (Italy). In this preliminary work, we focused on the possible link between insecure attachment, the use of coercion and violence in relationships, difficulty in controlling sexual fantasies and impulses, and sexual sensation-seeking. Furthermore, we investigated the personal perception of the risk factors that might lead to sexual recidivism.

We can advance some considerations according to the first results of the research. These hypotheses need to be confirmed by further analysis and administration. Most of our sample did not have sex talk within the family and confided in relatives or friends. In addition, they report perceived severe or abusive parental discipline during childhood. The absence of sex talk within the family might characterize a kind of deprivation in sexual development that can lead to future difficulties in the functional expression of sexuality. Children identify sexuality as something to be avoided or forbidden to talk about, internalize shame and fear, and stifle their fantasies and sexual needs with discomfort [2]. This discomfort can also lead to coercion, a lack of impulse control, and violence with one’s partners [11]. In addition, embarrassment for oneself and one’s sexualized body could lead some to develop a sense of discomfort in dealing with the adult world, leading to refuge in childhood and, therefore, the tendency to seek affection and sexual attention from children instead of adults.

Regarding the perception of severe or abusive parental discipline during childhood, it could derive from the perception of not feeling “loved” or understood by one’s reference figures, such as parents, and with the attribution of punishment—negative reinforcement—to one’s behavior being perceived as wrong. This conception is often associated with insecure attachment, anxiety to face decisions, and avoidance of relationships. Children, future adults, could reiterate that dynamic through self-directed or straight-aggressive acts [12,29]. The absence of affectivity, together with negligence in the children-parent relationship, can influence the affective and sexual development of the person.

Another factor linked to severe or abusive parental discipline is the experience of familial abuse, often associated with a future reiteration of the traumatic events or to the development of psychological or sexual issues [30]. When persons experience familial abuse, they could be unable to establish intimate contact with their parents. This experience can lead to investing in sexual or affective fantasies toward an “object” less anxiogenic, such as children. According to this point of view, specifically male sex offenders are attracted to children or women for what they represent. The dehumanization of the victim is functional to overcome the sense of post-traumatic impotence. Therefore, the real motivation of the abuser could be pseudo-sexual, as he aims at the solution of such experiences of deprivation [30].

In our sample, the percentage of participants who report suffering from physical, sexual, or psychological abuse during childhood is low. Furthermore, the number of participants who experienced a stable affective relationship (more than one year) is high. These could be positive factors to underline—even if we did not consider the quality of the relationship, but only the duration—and this might be a limit to highlight [31].

Regarding the possible connection between compulsive sexual behavior, sexual sensation-seeking, and attachment style, the data from the Pearson correlation show a positive association between difficulties in controlling sexual impulses and fantasies, use of violence in relationships, and the tendency to seek high-risk sexual situations. This aspect is an object of controversial results. Indeed, Efrati et al. [14] reported lower rates of CSBD among sex offenders in contrast with Sexaholics Anonymous (SA). Future research protocols should study this aspect.

In addition, from the analysis of the five factors of the Attachment Style Questionnaire (ASQ), we found a positive relationship among the “Relationships as Secondary scale” (which is part of the avoidant style), use of violence in relationships, and sexual sensation-seeking. The “Preoccupation with Relationships scale” positively correlates with sexual sensation-seeking and difficulty in controlling sexual impulses. Studies on sexual addiction and out-of-control sexual behavior have found that sexual compulsivity is associated with insecure attachment (anxious and avoidant) [32,33]. Compulsive sexuality could be a way to manage stressful events in subjects who have difficulties in emotion and behavioral regulation [33]. Moreover, research shows a strong association between attachment style and hypersexual and compulsive behavior [34,35,36], underlying the role of trauma as a mediator between insecure attachment and hypersexual behavior.

From the analysis of the personal perception of high-risk situations in sexual recidivism, most of the participants reported “anger and problems in managing it” under the negative emotion category. The sample shows high awareness of the emotion related to the offense and the perception of not being able to control it [37]. Regarding the positive emotions category, the participants identified the “sense of control on the behavior”, denoting a negative perception of control. Indeed, the participants think that the perception of a sense of control over sexual behavior can lead them to be less conscious of their actions. From this point of view, positive emotion means the “positive” evaluation of emotion that can translate into criminal or potentially risky conduct, amplifying impulsive actions [10]. In the thoughts about crimes category, the participants identified the answer “my behavior is correct”. They underlined the inability to evaluate the consequences of their actions, underestimating the impact of their offenses. This answer could be a common feature of sex offenders and might converge with moral disengagement mechanisms and cognitive distortions that characterize this category of offenders [37].

For instance, this thought may coincide with the concept of the thematic network [38]. According to the Judgment Model of Cognitive Distortions (JMCD), offenders can perceive themselves as being authorized to carry out the criminogenic action. The person believes themselves to be superior to others and tends to satisfy their needs unconditionally. This attitude leads to developing expectations of gratification from other people. Consequently, the offender believes that they deserve the satisfaction of their needs. In the neighborhood’s characteristics category, the participants answered: “Have contact with other offenders”, showing a high vulnerability and the presence of a background with a higher risk of relapse [4,18].

In the rehabilitation programs category, to the question “What reasons could negatively affect the success of possible rehabilitation programs?”, the participants answered: “Not to be confident in the treatment”. This lack of confidence in the treatment could derive from a perception of low investment by the institutions in sex offenders. In addition, it describes a marked resignation on the possibility of being integrated into society after imprisonment. The perception of disqualification is often the result of the social stigma and courtesy stigma that are part of the social context [39]. Sex offenders learn to internalize the stigma and do not believe in their resources [39].

Finally, to the question “What other situations or happenings could change your behavior?”, referring to the other general situations category, the participants answered: “Thinking about the future”. This answer describes the propensity of the sample to reflect on future possibilities and represents a positive investment for establishing rehabilitation programs. This category of offender feels the need to escape from the stigma of the crime. Thinking about the future for many of them coincides with the possibility of building a new life. The purpose is made difficult by two mechanisms: (1) social, cultural, and institutional disinvestment; (2) the offenders’ difficulty in being responsible for their conduct. These issues often lead to the instrumentalization of treatment and involve a decrease in both the social and individual confidence in the possibility of reducing relapse in these individuals [17].

## 5. Limits

This study presents some limits. First, the number of participants involved is small. Indeed, even if the target recruitment group represents a niche, the size of the sample could be expanded to involve several correctional facilities. Second, the data regarding family sex talk could be incomplete due to the lack of comparison with the geographic and cultural context of the general population. Indeed, is not possible to compare the percentage of family sex talk of sex offenders with those of men of the same age and area. Moreover, the measures used are self-reported; thus, the social desirability among the sample is higher and could represent a bias. In addition, only one self-report is validated in Italian, while the other scales used were not. This leads to an interpretative bias regarding the data collection. Finally, our sample involved a wide range of ages, with no differences among young adults, middle age, and senile age.

## 6. Conclusions and Clinical Implications

Despite the limited sample and the exclusive focus on the males, these preliminary data provided some suggestions and focus for future research and directions. A comparison between Italian women and men sex offenders should be conducted, with the aim to identify specific educational programs. In addition, our sample involved a wide range of ages. Future developments could compare different groups targeted for age, for instance, young adults, middle age, and senile age.

Furthermore, dividing and comparing groups according to their sexual offense is needed, to better identify the characteristics and specificities of the participants—child molester has a different profile from people who use violence against women-.

Moreover, we used adapted measurements that need to be implemented and validated to promote future research. Finally, should be promoted longitudinal studies on sexual recidivism to promulgate effective programs focused on relapse prevention.

Our findings could be useful in incrementing clinical programs aimed at social rehabilitation inside and outside correctional facilities, and in promoting sex talk and sexual education in juveniles to reduce the possible risk of developing compulsive sexuality, sexual sensation-seeking, and interpersonal violence in adulthood.

Moreover, participants reported some critical issues regarding the personal perception of high-risk situations linked to sexual relapse. These self-reported perceptions should be used in structuring specific and individualized programs aimed to reduce sexual recidivism and to increment self-awareness and critical revision of the offense. Data about self-reported perceptions of factors related to sexual recidivism could be also used by correctional facilities staff and treatment staff to integrate psychological and behavioral elements on justice-involved individuals.

## Figures and Tables

**Table 1 healthcare-11-01586-t001:** Socio-demographic questionnaire.

Childhood Family Communication about Sexuality and Education		
	Yes	No
Speaking of sexuality in the family	13.8% (N = 4)	86.2% (N = 25)
Confiding in parents about sexuality	24.1% (N = 7)	75.9% (N = 22)
Confiding in others about sexuality (friends or relatives)	55.2% (N = 16)	44.8% (N = 13)
Perception of severe or abusive parental discipline in childhood	55.2% (N = 16)	44.8% (N = 13)
Abuses in childhood		
Participants who suffered from sexual abuse	10.3% (N = 3)	
Participants who suffered from physical abuse	3.4% (N = 1)	
Participants who suffered from psychological abuse	10.3% (N = 3)	
Adult affective relationships		
Participants who experienced a stable affective relationship (more than one year)	96.6% (N = 28)	
Participants who did not experience a stable affective relationship (less than one year)	3.4% (N = 1)	

**Table 2 healthcare-11-01586-t002:** Correlation Between Sexual Sensation-seeking Scale, Compulsive Sexual Behavior Scale, and Attachment Style Questionnaire.

	1	2	3	4	5	6	7
1. SSSS_tot	-						
2. CSBI_control	0.562 **	-					
3. CSBI_violence	0.469 *	0.602 **	-				
4. ASQ_Confidence	−0.289	−0.167	0.147	-			
5. ASQ_ Discomfort with Closeness	0.297	0.270	0.277	0.328	-		
6. ASQ_ Relationships as Secondary	0.406 *	0.184	0.476 *	0.126	0.484 *	-	
7. ASQ_ Need for Approval	0.057	0.312	−0.053	−0.135	−0.002	0.090	-
8. ASQ_Preoccupation with Relationships	0.448 *	0.400 *	0.316	−0.037	0.422 *	0.529 **	0.506 **

*Note*: *p* < 0.05 * *p* < 0.01 **. SSSS: Sexual Sensation-seeking Scale; CSBI: Compulsive Sexual Behavior Inventory; ASQ: Attachment Style Questionnaire.

**Table 3 healthcare-11-01586-t003:** Personal evaluation of high-risk situations in sexual relapse.

Category	Question	Item	%
Negative emotions	Which of these negative emotions could lead you to reiterate the offense?	Anger and problems in managing it	24.1(N = 7)
Positive emotions	Which of these positive emotions could lead you to reiterate the offense?	Sense of control on the behavior	20.7(N = 6)
Thoughts about crimes	Which of these thoughts and actions could lead you to reiterate the offense?	My behavior is correct	24.1(N = 7)
Neighborhood’s characteristics	Which of these characteristics of the environment could lead you to reiterate the offense?	Have contact with other offenders	20.7(N = 6)
Rehabilitation programs	What reasons could negatively affect the success of a possible rehabilitation programs?	Not to be confident in the treatment	34.5 (N = 10)
Other general situations	What other situations or happenings could change your behavior?	Thinking about the future	51.7 (N = 15)

## Data Availability

Data will be available on request.

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
