# Peer review of "Family Attachment, Sexuality, and Sexual Recidivism in a Sample of Italian Sexual Offenders: Preliminary Results"

_healthcare, 2023, doi:10.3390/healthcare11111586_

Round 1

Reviewer 1 Report

This is an interesting study that presents some preliminary data drawn from interviews with a cohort of Italian male sexual offenders. The dimensions investigated (namely, family attachment, psychosexual correlates and sexual recidivism) are elegantly presented in the introduction section and coherently discussed in the discussion paragraph.  The scales used are appropriate and the final results may prompt future studies on the topic. However, I would like to make a number of critiques concerning the paper, which are not central issues in need to be revised, but rather limitations to the study that should be emphasized accordingly, better if listed in a separate paragraph:

- the sample of people investigated is rather small. Considering that the sexual offenders’ cohort is, relatively but unfortunately not always, a niche, this can be overlooked. However, the studies cited by the authors involved either a larger sample or, at least, a non-sexual offenders control group (or both), which could have greatly reinforced the results presented. I shall give an example: the authors reported that 86,2% of such males did not speak to their family about sexuality. This finding is impressive, but weak if taken alone, if one cannot exclude a possible influence of the geographic/cultural context (as said: how many males of the same age in Southern Lazio spoke with their family about sexuality?). Consequently, also the related discussion is flawed.

- 3 out of 4 of the scales used were not validated in Italian. It is unfortunately not possible to consider the language adaptation process as totally unbiased, hence this should be listed among the limitations.

- All the measures used are self-reported, thus the social desirability of the responses could not be excluded.

- My final suggestion is that the overall discussion would benefit from greater insight into the link between attachment style and hypersexual/compulsive behavior, given also the mediating role of trauma. Indeed, much data have been produced on the topic, (DOI: 10.1016/j.jad.2021.06.064; DOI: 10.1016/j.jad.2020.11.100; DOI: 10.1007/s40618-022-01798-3) but the authors unfortunately did not consider to include this into the discussion.

Minor issues: in table 1, spoking is likely to be “speaking”. Also, the concept of “severe education” is a bit inconsistent. What is severe and what is not?

Author Response

Thanks to the reviewer for the suggestions and feedback. We proceeded to add a section dedicated to the limits of the research and we enriched the discussions with the indicated references. Please see the attachment.

Reviewer 2 Report

The article examines a fascinating phenomenon: communication about sexuality and correlation between violence and sexual seeking. 

The issues discussed in the article are very important for understanding a very difficult problem like sexual abuse.

Methodology: please add information about interpretation of used scales and questionnaires. Regarding the selection of the participants, authors did not refer to the type of sampling or selection of the participants. Please mention what was the exclusion/inclusion criteria. There is no information about the recruitment of the  study sample. 

Results: there are some typos - f.e. "spoking" in Table 1. I also suggest to delete words "Percentage of the sample" in Table 1.

Discussion: Perhaps the authors could explain the limiations of the study.

Please add the conclusions section.

Finally, I recommend to discuss the conclusion of the research on the clinical implications of the results.

Author Response

Thanks to the reviewer for the suggestions and feedback. We proceeded to
enrich the methodology and the discussions section. We also added limits
of the study and the clinical implications of the results. Please see the attachment

Round 2

Reviewer 1 Report

Nice amendments for a nice study. 
Minor error: line 352 "incremented"; capital letters for correlational facilities are not needed.

Author Response

Thanks to the reviewer for the comments.

We revised line 352 "incremented" and capital letters for correlational facilities.